# Development of Films from Spent Coffee Grounds’ Polysaccharides Crosslinked with Calcium Ions and 1,4-Phenylenediboronic Acid: A Comparative Analysis of Film Properties and Biodegradability

**DOI:** 10.3390/foods12132520

**Published:** 2023-06-28

**Authors:** Michelle J. P. A. Batista, M. Betânia F. Marques, Adriana S. Franca, Leandro S. Oliveira

**Affiliations:** 1PPGCA, Universidade Federal de Minas Gerais, Av. Antônio Carlos, 6627, Belo Horizonte 31270-901, MG, Brazil; michelle.jp.azevedo@gmail.com (M.J.P.A.B.); adriana@demec.ufmg.br (A.S.F.); 2DQ, Universidade Federal de Minas Gerais, Av. Antônio Carlos, 6627, Belo Horizonte 31270-901, MG, Brazil; 3DEMEC, Universidade Federal de Minas Gerais, Av. Antônio Carlos, 6627, Belo Horizonte 31270-901, MG, Brazil

**Keywords:** bioplastics, by-products, galactomannan, calcium ion, 1,4-phenylenebisboronic acid, crosslinking, ssNMR, polysaccharide, zinc chloride

## Abstract

Most polymeric materials are synthetic and derived from petroleum, hence they accumulate in landfills or the ocean, and recent studies have focused on alternatives to replace them with biodegradable materials from renewable sources. Biodegradable wastes from food and agroindustry, such as spent coffee grounds (SCGs), are annually discarded on a large scale and are rich in organic compounds, such as polysaccharides, that could be used as precursors to produce films. Around 6.5 million tons of SCGs are discarded every year, generating an environmental problem around the world. Therefore, it was the aim of this work to develop films from the SCGs polysaccharide fraction, which is comprised of cellulose, galactomannans and arabinogalactans. Two types of crosslinking were performed: the first forming coordination bonds of calcium ions with polysaccharides; and the second through covalent bonds with 1,4-phenylenediboronic acid (PDBA). The films with Ca^2+^ ions exhibited a greater barrier to water vapor with a reduction of 44% of water permeability vapor and 26% greater tensile strength than the control film (without crosslinkers). Films crosslinked with PDBA presented 55–81% higher moisture contents, 85–125% greater permeability to water vapor and 67–150% larger elongations at break than the films with Ca^2+^ ions. Film biodegradability was demonstrated to be affected by the crosslinking density, with the higher the crosslinking density, the longer the time for the film to fully biodegrade. The results are promising and suggest that future research should focus on enhancing the properties of these films to expand the range of possible applications.

## 1. Introduction

Since the mid-20th century, society has been increasingly using polymeric materials, mostly due to their versatility, stability and low cost [1]. However, single-use materials, particularly packaging, have contributed significantly to plastic waste and their disposal has been largely inadequate. Improperly discarded, plastic waste ends up either in landfills or waterways, ultimately reaching the oceans in the form of microplastics, which have even been found in human breastmilk [2,3]. Predictive models estimate that 5 to 13 million tons of plastics are deposited in the ocean annually, which can take from 500 to 1000 years to be degraded [2]. However, polymeric materials, such as plastics, are widely used due to the necessity to maintain the integrity of a wide variety of products during storage, handling and transportation. Furthermore, these materials have functions associated with information, containment and marketing. Regarding food, these materials are used as packaging to secure the quality and safety of products, extending shelf life by hindering interaction with microorganisms, water vapor, oxygen, off-flavors and losses of desirable compounds [4,5,6].The development of biodegradable polymeric films from agri-food waste is a suitable alternative to reduce the production of conventional synthetic packaging materials derived from fossil resources such as petroleum [7,8]. Such biopolymeric materials are increasingly regarded as promising green substitutes of non-biodegradable plastic materials, offering environmentally friendly and sustainable solutions [4]. Polysaccharides are the most abundant biopolymers in nature and are a good alternative for producing biomaterials, since they are of renewable and recurrent sources, biocompatible, non-toxic and biodegradable, with the latter trait being highly desirable in regard to the mitigation of negative environmental issues caused by non-biodegradable polymers from fossil sources [7]. Furthermore, polysaccharides are easily processed into various forms, such as capsules, films and fibers [8], and can be obtained from agricultural and food production wastes, such as spent coffee grounds (SCGs).

Coffee is the second most consumed beverage in the world, the first is water. In 2020, coffee production reached nearly 176 million 60 kg bags, resulting in the generation of approximately 6.5 million tons of spent coffee grounds. SCGs are the wastes generated from the solid–liquid extraction process involving water and ground coffee beans for the industrial production of instant and soluble coffee and the preparation of household and commercial brews [9]. SCGs are food wastes rich in organic compounds such as polysaccharides, phenolics, proteins and fatty acids. About 66% (m/m, db) of SCGs are polysaccharides, which are comprised of galactomannans (50%), arabinogalactans (25%) and cellulose (25%) [10]. Galactomannan (GM) consists of D-mannopyranose residues linked to each other by β-(1→4) bonds in the main chain, with side chains of individual residues of D-galactopyranose (D-Galp) linked to the main chain through α-(1→6) bonds [11,12]. The mannose/galactose ratio (M/G) is one of the main characteristics used to differentiate galactomannans of different sources [13,14]. The most commonly used galactomannans in the food and pharmaceutical industries are guar gum, tara gum and locust bean gum, which present M/G ratios of 1.6–1.8, 3.0 and 3.9–4.0, respectively [15], whereas coffee galactomannans present comparatively low degrees of substitution, in the range of 14:1 to 30:1 [16,17]. Such a polymeric structure favors the formation of crystalline regions with extensive inter- and intramolecular interactions in the portions of the molecules with no substitutions, hindering solubilization, similar to what happens with cellulose crystalline regions [18]. The mannose/ galactose ratio, degree of substitution and galactose distribution pattern notably influence the film-forming properties of galactomannans. The chemical structure of coffee galactomannans allows for denser packings than usual commercial galactomannans allow [13,19,20].

Coffee arabinogalactans are Type II, which means they are highly branched. The main chain is composed of D-Galp residues linked by β-(1→3) and with side chains formed by units of D-Galp linked to the main chain by β-(1→6) bonds, its side chains present branches of residues of α-L-arabinose, α-L-rhamnose and α-D-glucuronic acid. Coffee arabinogalactans are predominantly bound to protein fragments (≈4%), forming glycoproteins known as arabinogalactan-proteins [21,22]. Cellulose, a homopolymer comprised of glucose units with β-(1→4) linkages, is the most abundant polymer in nature and makes up 25% of the polysaccharides in SCGs [23,24]. The β configuration allows for the formation of long and linear chains, enabling the organization of fibrils by hydrogen bonds between the polymeric chains, with the molecules interacting parallel to each other [25]. This configuration is responsible for the high tensile strength cellulose fibers present, as is also the case with mannans (or galactomannans with a low degree of galactose residue substitution). The intramolecular hydrogen bonds of galactomannans are stronger than the hydrogen bonds in cellulose, and this difference is due to the presence of hydroxyl pairs in the cis conformation for carbons C2 and C3 in the mannose residues of galactomannans, instead of the trans configuration that occurs in the glucose units of cellulose [26]. Thus, the two main polysaccharides in spent coffee grounds, cellulose and galactomannans, are natural sources to produce biodegradable films or coatings. For this reason, the agri-food waste SCGs can be upcycled into a diversity of products, including biopolymeric films, composites and biomaterials [19,27,28].

Previous studies have shown the potential of using SCGs to produce biopolymeric films [19,29]. However, the physical, structural, thermal and mechanical properties of biopolymer films are often inadequate for market needs due to inferior performance in comparison to conventional petroleum-derived packaging materials [4,30]. Polysaccharide-based films typically exhibit favorable barrier properties against oxygen under low and moderate relative humidity conditions, while also demonstrating relatively acceptable mechanical strength [5]. However, these films often demonstrate limited moisture and water vapor barrier capabilities due to their inherent hydrophilic nature. Their moisture sensitivity is one main reason for such films exhibiting inadequate characteristics compared to well-established commercial polymers because it can affect mechanical, thermal and barrier properties.

Thus, it was the aim of this work to verify the feasibility of employing crosslinking agents to enhance the mechanical, thermal and water vapor barrier properties of SCGs-based biopolymeric films. Two types of crosslinking were investigated: the first forming a coordination complex of calcium ions with polysaccharides; and the second one using 1,4-phenylenediboronic acid (PBDA), a compound with two boronic groups that form covalent ester bonds with the diols in monosaccharides [16]. The rationale for studying two different types of crosslinking is as follows. Calcium ions form a coordination complex with the hydroxyl groups of the monosaccharides, closely approximating the polymeric chains of the matrix, which allows for the enhancement of the water vapor barrier and mechanical properties of the films. Calcium ions are cost-effective and, furthermore, establish bonds that can be reversible, which may be useful for making biodegradable materials [31,32]. In addition, information in the literature about using divalent ions with neutral polysaccharides (e.g., galactomannans) for the development of films is scarce. On the other hand, the crosslinking by PBDA can form reversible covalent bonds with the diol groups of monosaccharides, but the expected interactive distance between polymeric chains would not be as short as with calcium ions [33,34]. It is noteworthy to mention that the boronic groups in PBDA are at opposite ends of the molecule and have the same pKa (8,3), which favors the interaction of each boronic group with diols of distinct polysaccharide molecules. Furthermore, the aromatic ring present in the molecule structure of PBDA certainly provides greater stability to the ester bonds of its boronic groups with the monosaccharide cis-diol groups.

## 2. Materials and Methods

### 2.1. Materials

SCGs were collected from a coffee shop in Belo Horizonte, Brazil. The samples were dried in a convective oven at 105 ± 5 °C, for 16 h. After drying, SCGs were ground (0.50 < D < 0.84 mm) and stored in polypropylene packages at room temperature, prior to further analysis.

### 2.2. Removal of Phenolics from Spent Coffee Grounds

The removal of phenolics from SCGs was performed according to the methodology reported by Batista et al. [19]. A total of 40 g of spent coffee grounds were added to 200 mL of 35% (*v*/*v*) hydrogen peroxide solution and the solution pH was adjusted to 11.5 by adding a 40% (*m*/*v*) NaOH solution. The suspension was agitated at 120 rpm for 18 h at 25 °C in an orbital incubator. The mixture was neutralized with 10% (*v*/*v*) acetic acid solution and then filtered. The filtration residue was placed in a convective oven for 16 h at 55 °C. The samples were ground, sieved through a 20-mesh sieve (Tyler series) and stored for further analysis. The samples resulting from the alkaline hydrogen peroxide treatment were labelled SCGs-AHP.

#### 2.2.1. Composition Analysis of SCGs and SCGs-AHP

The proximate compositions of SCGs and SCGs-AHP were assessed using AOAC methods [35]. Moisture content was determined through oven drying at 105 °C until a constant weight was achieved. The Soxhlet method with petroleum ether was employed to measure crude fat. Ash content was quantified by subjecting the samples to burning at 600 °C until white ashes were obtained. The micro-Kjeldahl nitrogen method was utilized to estimate protein content, with a conversion factor of 6.25 applied to convert nitrogen into protein content. TAPPI 222 om-02 [36] was employed to determine the content of condensed phenolic compounds. Carbohydrate content was calculated by employing the difference method.

#### 2.2.2. Sugar Analysis of SCGs and SCGs-AHP

The monosaccharides in the SCGs and SCGs-AHP samples were identified following the method described by Sluiter et al. [37]. For this purpose, the samples were hydrolyzed by sulfuric acid. A calibration curve with five points ranging from 0.1 to 4.0 mg/mL was determined for each standard. Each sample was prepared in triplicate. The analysis of the samples was performed using an HPLC system (Shimadzu LC-20AP) equipped with a Supelcogel C-610H column (30 cm × 7.8 mm). The mobile phase consisted of 5 mM sulfuric acid, and the flow rate was set at 0.6 mL/min. The column temperature was maintained at 60 °C, and a volume of 20 μL was injected. A refractive index detector was used for sample readings.

### 2.3. Film Production

The preparation of films was based on the Loeb–Sourirajan technique involving precipitation of a casting solution by immersion in a nonsolvent bath as described by Batista et al. [19], with modifications. Firstly, 5% (*w*/*w*) of the SCGs-AHP sample was added to a 67% (*w*/*w*) ZnCl_2_ solution (39.38 g of ZnCl_2_ dissolved in 16 mL of distilled water and heated to 65 °C in a water bath under agitation for 10 min). A paste was prepared by adding 2.94 g of SCGs-AHP to 3.5 mL distilled water. The ZnCl_2_ solution was added to the paste and stirred at 65 °C for 90 min. Then, the filmogenic solution was cast into a silicone dish (20 × 20 cm). Subsequently, the support with film solution was immersed in 250 mL of anhydrous ethanol at 8 °C for 30 min to coagulate the film matrix. To remove excess zinc chloride, the coagulated matrix was immersed in 400 mL of water distilled at 75 °C for 2 min. The film was then soaked in glycerol (10% *v*/*v*) for 15 min, followed by drying at 21 °C for 48 h under a relative humidity of around 50%. The resulting film was considered the control film and termed CF. All films were stored in low-density polyethene plastic bags with hermetic closure and kept at room temperature for further analysis.

#### 2.3.1. Crosslinking with Calcium Ions

To produce films crosslinked with calcium ions, the methodologies described by Pavlath et al. [31] and Rhim [38] were used. To produce the films, the methodology for the control film was used and the dried films were immersed in a CaCl_2_ solution with concentration and time determined according to Table 1. Subsequently, the films were dried at 21 °C under a relative humidity of 50% for 48 hours.

#### 2.3.2. Crosslinking with 1,4-Phenylenediboronic Acid (PDBA)

The methodology used for the development of films with PDBA was based on studies by Thombare et al. [33] with modifications. First, films were prepared using 1,4-phenylenediboronic acid at concentrations of 2.5, 3.5, 5.0 and 7.5% (m/m of SCGs-HPA). The 7.5% PDBA concentration saturated the solution, producing films with undissolved reagent particles in the matrix, thus this film was not further analyzed. PDBA films were developed using the CF sample methodology described in Section 2.3 with some modifications as follows. After SCGs-HPA was dissolved in a zinc chloride solution, forming a homogeneous film-forming solution, an aqueous suspension containing PDBA was added to this solution. Before adding the PDBA suspension to the filmogenic solution, the suspension was vortexed, placed in an ultrasonic bath at 50 °C for 20 min, and vortexed again. Then, the PDBA suspension was added to the filmogenic solution and stirred for 30 min. The resulting suspension was poured into a glass support and the same steps used for the preparation of the control film were carried out. The films prepared thereof were termed F2.5, F3.5 and F5 and the treatments are summarized in Table 2.

### 2.4. Film Characterization

Prior to the characterization of the prepared films, the samples were conditioned at 23 °C and 50% relative humidity for at least 40 h [24].

#### 2.4.1. Fourier-Transform Infrared (FTIR) Spectroscopy

All samples were analyzed by a Shimadzu IRAffinity-1 FT IR Spectrophotometer (Shimadzu, Japan) with a DLATGS detector (Deuterated Triglycine Sulfate Doped with L-Alanine). Scan range was set at 4000−500 cm^−1^ with 20 scans at 4 cm^−1^ resolution. Spectra were obtained with an Attenuated Total Reflectance (ATR) accessory with a ZnSe crystal, at 20 °C and 50% relative humidity. The spectra were corrected for CO_2_ and spectral noises using the IR-Solution software (version 1.50—Shimadzu).

#### 2.4.2. Thermal Analysis

Thermogravimetric analysis (TGA) of the prepared films was carried out using a DTG60 Shimadzu equipment (Shimadzu, Japan). An open alumina crucible was filled with 1.5 to 3.0 milligrams of the sample. Measurements were carried out between 25 and 600 °C with a linear temperature increase of 10 °C min^−1^. The experiments were run under N_2_ gas flow of 50 mL min^−1^ [24]. The Ta60 software version 2.21 calculated sample weight loss, derivative thermogravimetric curve (DTG) and the decomposition temperature of the samples.

#### 2.4.3. Nuclear Magnetic Resonance Spectroscopy (NMR)

For the nuclear magnetic resonance analysis, the films were powdered and packed into 4 mm zirconia rotors. All the experiments were performed at 25 °C with a 5 kHz spinning rate. The equipment used to collect the solid-state NMR spectra was a Bruker Avance III 400 MHz equipped with a standard Bruker 4 mm magic-angle spinning (MAS) probe, operating at 100.57 MHz for a 13C nucleus. The 13C NMR spectra were acquired with cross-polarization and total sideband suppression (CPTOSS) pulse sequence, collecting 2048 scans with a recycle delay of 3 s, 2 ms of contact time and 34 ms of acquisition time.

#### 2.4.4. Moisture Content

The moisture content was determined using standard gravimetric methods of analysis [35]. A total of 0.5 g of each film was dried in an oven at 105 °C for 24 h. The dried sample was placed in a desiccator until it reached room temperature, and then weighed.

#### 2.4.5. Stability in Acidic and Alkaline Solutions

For the evaluation of the stability of films in acidic, neutral and alkaline solutions, the methodology described by Medina Jaramillo et al. [39] was employed with some modifications. First, specimens of 16 ± 0.1 mm diameter were immersed in 10 mL of solutions of distinct pHs: a solution of hydrochloric acid at pH 3; a solution of sodium hydroxide at pH 12; distilled water at pH 7. The samples were kept immersed in the solutions for 10 days at 21 °C, after which the measurements of the samples’ diameters were performed using a caliper with a resolution of ±0.01 mm.

#### 2.4.6. Water Solubility

The evaluation of film water solubility was based on the method described by Antoniou et al. [40] with a few modifications. Initially, the samples were cut into squares (2 × 2 cm), dried and weighed. The specimens were immersed in 30 mL of distilled water at 25 °C under agitation for 24 h. The material was filtered, and the insoluble part dried at 105 °C for 24 h. Water solubility (*WS*%) was calculated as follows:(1)$$WS\left(\%\right)=\frac{w_{i}-w_{f}}{w_{i}}\times 100$$
where $w_{i}$ represents the initial weight of the dry sample and $w_{f}$ represents the weight of the dry material that remains insoluble.

#### 2.4.7. Water Vapor Permeability (WVP)

Water vapor permeability (WVP) of the films was determined according to ASTM E96 [41] and Do Lago et al. [42]. Glass vials of 16 mL, 71 mm in height and a lid with an opening of 11 mm as internal diameter were used to determine the *WVP* of the samples. The volume of each vial was filled with ¾ of silica previously dried at 150 °C for 24 h. The samples were cut and fitted between the flask and the cap. To ensure that the flask was sealed, silicone grease was used on the edges of the samples to force water vapor to permeate only through the specimens. Initially, the vials were weighed and placed in a desiccator containing a saturated sodium chloride solution at 21 °C. Then, the flasks were weighed for 7 consecutive days. All tests were carried out in four replicates, and the water vapor permeability was calculated according to:(2)$$WVP=\frac{G}{t\;A}\times \frac{l}{S\left(R_{1}-R_{2}\right)}$$
where *G*/*t* is the slope of the weight gain (g) versus time *t* (h) curve; *A* (m^2^) is the test area of the film specimen; *l* (mm) is the mean thickness of the specimen; ($R_{1}-R_{2}$) represent the difference in relative humidity between the interior of the vial and the desiccator; and *S* is water vapor pressure (kPa) at 21 °C.

#### 2.4.8. Morphological Analysis

To evaluate the morphology of the films, a Quanta 200 FEG (FEI) scanning electron microscope was used. The samples were metalized with carbon tape. Acceleration voltages of 2 and 20 KV were used to obtain images of the surface and film interface morphology, respectively.

#### 2.4.9. Biodegradability

Biodegradability analysis of the films was carried out in a soil prepared according to ASTM G160 [43], with three components mixed in equal parts: fertile soil with low clay content, manure, and sand. The pH and soil moisture content were monitored and maintained between 6.5 and 7.5, and between 20 and 30%, respectively. After three months of monitoring the soil, the film samples were buried in it to evaluate their biodegradability. The biodegradability test was carried out according to the methodology described by Medina Jaramillo et al. [39]. Containers with a depth of 12.5 cm were filled with soil up to 4 cm in height. The test specimens (2 × 2 cm) were buried in the soil at a depth of approximately 1 to 2 cm in an aerobic condition. The containers were kept at room temperature and the soil moisture was regularly replenished by spraying water. The samples were removed and analyzed after 5, 10, 30, 90 and 180 days.

#### 2.4.10. Mechanical Properties

The tensile strength of the prepared films was determined using the ASTM D882 standard [44] with some modifications. The tests were carried out using a Stable Micro Systems texture meter (TAXTPLUS) equipped with a 25 N load cell. The initial grip separation was 40 mm and the crosshead speed used was set at 2 mm s^−1^. Five test specimens of 80 × 10 mm were used for each analysis. Film thickness was measured using a Mitutoyo micrometer (No 103 137) at three different randomly selected points on each test specimen to calculate the tensile strength (TS). Films were tested with their moisture contents as determined by the gravimetric procedure described in Section 2.4.4.

## 3. Results and Discussion

### 3.1. Characterization of SCGs and SCGs-AHP

#### 3.1.1. Composition Analysis

The average yield of phenolics-depleted SCGs was 56.35 ± 2.68%, with the alkaline peroxide treatment efficiently removing approximately 76% of the phenolics in SCGs (Table 3). Aside from the free chlorogenic acids, the treatment with hydrogen peroxide also partially removed melanoidins and condensed phenolics from the SCGs matrix. Hydrogen peroxide decomposes into *HOO*^−^ anions and into superoxide anions and hydroxyl radicals, which react with aromatic structures, double bonds and carbonyls [45,46]. Such chemical functionalities are present in condensed phenolics and melanoidins. Coffee melanoidins contain chlorogenic acids covalently linked to their structure as a result of Maillard reactions. The reactive oxygen species resulting from the hydrogen peroxide decomposition can also react with polysaccharides (mostly with arabinogalactans and galactomannans in SCGs) causing mass losses in such fractions [47]. Therefore, the efficiency of the treatment with alkaline hydrogen peroxide is a relevant aspect to be evaluated since it directly affects the content of polysaccharides, which are the constituents of interest for the preparation of films. An analysis of the data regarding H_2_O_2_ treatment allows the resultant loss of carbohydrates to be estimated at 23%. However, this mass loss was lower than the losses in the other fractions (i.e., proteins, lipids, phenolics, moisture and ash), thus increasing the polysaccharide content of the treated product (63.41%) in regard to the original untreated SCGs matrix (46.31%).

Hydrogen-peroxide-treated SCGs presented a lower content of proteins, a similar content of lipids, and a higher content of ash than untreated SCGs. The reduction in protein content is herein attributed to a reduction in protein conformational stability promoted by the hydrogen peroxide and consequent hydrolysis caused by sodium hydroxide [48]. The lipids in the SCGs matrix might have been slightly oxidized by the reactive oxygen species generated by the hydrogen peroxide decomposition [49].

#### 3.1.2. Sugar Analysis

The monosaccharide composition of SCGs is presented in Table 4. According to the literature, galactomannans comprise approximately 50% (m/m, d.b.) of the polysaccharides in SCGs [50,51], and the results herein obtained corroborate that, with mannose being the predominant monosaccharide. Recall that SCGs galactomannans have mannose: galactose ratios in the vicinity of 30:1. The majority of glucose is from cellulose (β-1→4-glucopyranose polymer), arabinose comes from arabinogalactans and galactose is present both in galactomannans and arabinogalactans, as side and backbone chains, respectively [21,52,53]. The increases in contents of glucose and galactose are attributed mostly to the relative losses of arabinose caused by the action of the reactive oxygen species generated by the hydrogen peroxide decomposition. 

The monosaccharide composition of SCGs-AHP was similar to that of SCGs, with major differences related to the reduction in arabinose molar content from 19% in SCGs to 13% in SCGs-AHP and to the increase in glucose content from 26% in SCGs to approximately 31% in SCGs-AHP. Therefore, it can be readily concluded that most of the cellulose molecules were preserved during the treatment and arabinose from the arabinogalactans was the monosaccharide most susceptible to degradation [54].

### 3.2. Characterization of Films

#### 3.2.1. Fourier-Transform Infrared (FTIR) Spectroscopy

The functional groups and chemical bonds of the films developed in this study were analyzed using mid-infrared spectroscopy. The spectra of the film samples crosslinked with Ca^2+^ ions are presented in Figure 1. F511 films were not analyzed by FTIR because they were extremely brittle, rendering their handling virtually impractical. The film crumbled into significantly small pieces upon handling, therefore, making the analysis performed with the ATR accessory unfeasible, since the accessory has a rather large surface area that needs to be completely covered for the analysis to be suitably carried out.

The 3600–3000 cm^−1^ band is related to the stretching vibrations of O−H groups and generally presents a wide band due to inter- and intramolecular hydrogen bonds [55]. This region was attributed to the presence of water and polysaccharides [19,56]. The F411 film spectrum showed greater intensity in this region than the other samples’ spectra. The molecular force between the polymer chains may have been reduced by the presence of Ca^2+^ ions that acted as plasticizers, resulting in a film with greater water absorption [57].

Unlike the other samples, F411 showed an intense peak at 1641 cm^−1^, which may also be associated with the presence of water bound to the polymer matrix [58,59], corroborating the hypothesis that the Ca^2+^ ions along with water molecules promoted the plasticizing effect in F411. The interaction between the ions and polysaccharides is responsible for the crosslinked bonds between the polymer chains, however, if the moisture content in the films increases, it may be an indication that the ions attracted water molecules, causing the plasticizing effect on the polymer matrix [58,60]. Additionally, the spectrum of the F411 also presented more defined peaks with increased intensities at 2922 and 2852 cm^−1^, which were attributed to the asymmetrical and symmetrical stretches of the CH bonds in the methyl (−CH_3_) groups, respectively. The higher intensity in these peaks in the F411 spectrum may have occurred due to the increased mobility of the polymers, while the low intensity of these peaks in F311 suggests that the bonds between the polymer chains are stronger and the structure more rigid [29].

The 1575–1352 cm^−1^ band may involve the overlapping of peaks related to the vibration of various bonds attributed to the different polymers in the matrix, i.e., cellulose, hemicellulose and bonded phenolics. Similarly, the peak at 1434 cm^−1^, which can be attributed to the vibration in the O−H plane and is associated with the presence of holocellulose and phenolics [32], is intensified in the presence of Zn^2+^ and Ca^2+^ ions which interact with the hydroxyls in the C−O−H group [29,32].

The 1300–900 cm^−1^ region is the fingerprint region and is characterized by the presence of complex peaks and bands, such as the 1150–950 cm^−1^ band. This band is indicative of vibrations of the C−O bond in the C−O−H functional group, as well as symmetrical and asymmetrical vibrations of C−O−C bonds, typical of hemicellulose and cellulose [32,58]. In the 1150–950 cm^−1^ region, the CF sample showed a peak at 1027 cm^−1^, whereas the other films exhibited two peaks, at 1070 and 1012 cm^−1^. The peak at 1027 cm^−1^ in CF is associated with asymmetrical stretching vibrations of C−O bonds in C−O−C, and the peak at 1070 cm^−1^ is linked to vibrations in the C−O bond of alcohols [61]. The presence of these two peaks in the 1150–950 cm^−1^ region suggests an interaction between the functional groups of polysaccharides and Ca^2+^ ions. The shift of the peak at 1027 cm^−1^ to a lower frequency in the infrared spectrum indicates that the oxygen atom of the primary alcohol C−O−C ring of the monosaccharide moiety interacted with Ca^2+^ [58].

The peaks at 818 and 875 cm^−1^ are present in all films, although with different intensities, and are associated to α and β-glycosidic bonds, being indicators of the presence of α-D-galactopyranose and β-D-mannopyranose, respectively, both constituent monosaccharides of galactomannans [14,62]. The peak at 926 cm^−1^ present in the spectrum of the control film is related to glycosidic bonds in cellulose [63], its shift in the spectra of the other samples indicates that the Ca^2+^ ions have interacted with the polymer chains of cellulose.

Regarding the spectra of the films with crosslinking with PDBA, there are only a few distinguishing peaks regarding the control film (CF), as shown in Figure 2.

The peak at 1742 cm^−1^ present in the CF spectrum is also present in F2.5, although with reduced intensity, and it is absent in the spectra of samples F3.5 and F5. This suggests that the C=O functional group of ester and carboxylic groups, which can be attributed to the presence of acetyl groups esterified to galactomannans and uronic acids in arabinogalactans, respectively, interacted with PDBA through covalent bonds.

The reduction in intensity that occurred to the peak at 1647–1641 cm^−1^ indicates that hydroxyls of the polysaccharides may interrupt interaction with the water molecules and form bonds with the PDBA hydroxyls [33,64]. In addition, these alterations in peak intensity might be associated with the interaction of PDBA with the side-chain amine groups of the residual proteins linked to arabinogalactans (i.e., −N−B− bond). Additionally, the disappearance of the 1457 cm^−1^ and 1150 cm^−1^ bands, which are attributed to the stretching of the C-O-C in the pyranose ring and to C-H bonds, indicate the interaction of the polysaccharides with PDBA.

Furthermore, PDBA molecules may have interacted with bonded phenolics, which remained in the matrix even after treatment with AHP. The bands at 1645 cm^−1^ and 1528 cm^−1^ can be associated with the vibrations of the carbonyl group (C=O) stretching and the C=C vibrations of aromatic rings, respectively, both being attributed to the presence of bonded phenolics [65]. Additionally, the peak at 1452 cm^−1^ may also be related to the deformation of the C-H bond in groups in polyphenolics [63].

#### 3.2.2. Thermal Analysis

TGA was used as a technique to verify the crosslink density of the films. Increased crosslink density generally results in greater thermal stability for the polymeric material, which depends not only on the inherent characteristics of the matrix but also on the interactions between the macromolecules [31,43,44]. 

The TGA curves of CaCl_2_-crosslinked samples exhibited distinct behaviors, reflecting the impact of both the concentration and the exposure time of the films to the calcium chloride solution bath. Additionally, it can be noted that both the time and concentration had a negative effect on the thermal stability of the material within the evaluated range. As the exposure time and concentration increased, the value of the main peak temperature of the DTG decreased (Figure 3).

In general, the mass loss of the first stage in the range of 30–120 °C, which is present with some variations in all samples, corresponds to the evaporation of water molecules adsorbed in the material [24,66]. 

Hemicellulose and cellulose decomposition temperatures generally occur between 250 and 300 °C and 300 and 350 °C, respectively, with some variations [67,68]. However, in film analysis, two factors that can decrease the thermal stability of the polysaccharides must be considered. Firstly, the dissolution of the polymers in the film-forming solution affects the intermolecular bonds and crystallinity of the polysaccharides, resulting in a reorganization of the polymers into a configuration that is different from the original one. Secondly, during the preparation of the film, the zinc chloride solution may partially degrade the polymers present in the matrix, affecting its thermal stability [24].

As expected, CF is the most distinct of all since it was not immersed in the CaCl_2_ bath. Its curve displayed two prominent peaks in the DTG: the first occurred at 180 °C and the second at 295 °C, with a mass loss of around 30% in each peak. Both peaks are attributed to the decomposition and depolymerization of the polysaccharides present in the SCGs. The first peak is likely associated with the degradation of arabinogalactans, which have lower thermal stability compared to the main polysaccharides in the matrix, while the second peak is related to the degradation of galactomannans and cellulose [52,62].

The F411 sample exhibited a distinct thermal behavior in comparison to the other films. It presented the highest mass loss during the first stage, around 25%, which is consistent with its FTIR spectrum presented in Section 3.1 (Figure 1). These results indicated that F411 had the highest water content, which may be related to the higher percentage of Ca^2+^ ions in the matrix that were not involved in crosslinking. This sample showed an intense degradation of hemicellulose with a peak at 220 °C, which was the lowest temperature of decomposition of these compounds among all CaCl_2_ treatments. Therefore, it can be inferred that the Ca^2+^ ions and free water molecules in the matrix acted as plasticizers in the films, reducing intermolecular forces [58,69].

Samples F311 and F511 showed similar thermal behavior, as can be noted from the thermograms in Figure 3a. Both treatments also showed the peaks with the highest temperatures in the DTG curves, as shown in Figure 3b. These samples exhibited the lowest mass loss in the first stage and thus, probably the lowest water content, which is a consequence of decreasing free hydroxyl groups [70]. It was consistent with their rigid and brittle structure, as well as an opaque appearance (see Appendix A). This may suggest that there is a strong interaction between the polymer chains of the materials due to the high crosslink density [58].

The additional stage of decomposition observed in TGA curves of samples F311 and F511 compared to the other CaCl_2_-crosslinked samples is likely due to the interactions between some chains of polymers promoted by Ca^2+^ ions present in the matrix. The third peak in the DTG of samples F311 and F511 indicates that the formed crosslinks require more energy to be degraded.

Regarding the results of the thermal analysis of the PDBA-crosslinked films, it was observed that the TGA curves presented thermal behavior similar to that of the control film, as can be seen in Figure 4. In addition to the similarity between the degradation profiles of crosslinked films with PDBA and the control film, F3.5 showed an additional degradation stage between 326 °C and 429 °C with a peak in the DTG at 371 °C and a mass loss of 11%. Thus, the fourth degradation event present in the TGA curve of F3.5 is an indication of the formation of crosslinked bonds between the polymeric chains and PDBA. It may be related to the degradation of the links between PDBA, and the hydroxyl groups present in the polysaccharide matrix of the films. Similar thermal behavior was observed by Thombare et al. [33], who added borax as a crosslinking agent in guar gum hydrogels and observed the emergence of an additional event with a degradation range of 330 °C to 508 °C before the complete combustion region.

However, samples F2.5 and F5 did not show this additional stage, which suggests a scarce formation of crosslinked bonds between the polymeric chains and PDBA. The crosslinking density in both films was probably lower than the percentage formed in F3.5, making such an event imperceptible in the TGA and DTG curves.

#### 3.2.3. Nuclear Magnetic Resonance Spectroscopy (NMR)

Solid-state NMR allows for the analysis of polymeric materials, such as the developed films, without the need for any treatment, preserving the original structure of the samples and enabling comparison between the different developed films.

The peaks with the highest intensities (Figure 5) are related to the carbons in the polysaccharide chains of the coffee waste, namely cellulose, arabinogalactans and galactomannans. In general, the signals located from δ 58 to δ 66 ppm are related to C6, from δ 66 to δ 79 ppm to carbons C2, C3 and C5, from δ 79 to δ 92 ppm to C4, and from δ 96 to δ 108 ppm to the anomeric carbon C1 [71,72,73].

Comparing the spectra of CF with the spectra of samples F111, F211 and F411 in Figure 5, it can be observed that the latter have a lower resolution compared to the former, probably due to the reduction in the mobility of the polymer matrix promoted by the crosslinking by Ca^2+^ [74,75]. In addition, the peak intensities decrease in the spectra of CaCl_2_-crosslinked samples indicating the degree of affinity between the ions and the polymers present in the matrix. Therefore, the results suggest that F211 is the treatment that allowed for a better formation of crosslinks in the sample, reducing the incompatibility between the different compounds in the matrix [76]. The intensity change and chemical shift to higher or lower fields of carbon-13 associated with the monosaccharides indicates that calcium ions affected the bonds of each carbon of the polysaccharide chain, suggesting that even though these units are not charged, since they are predominantly neutral polysaccharides, residues mainly composed of mannose, glucose and galactose showed a high tendency to interact with the added ions in the matrix [74].

Regarding the spectra of the PDBA-crosslinked samples, it can be observed that they have a higher resolution, probably due to the greater mobility of the chains present in the samples when compared to CF (Figure 6) and the other films [75].

In addition, the NMR spectra suggest that the PDBA molecules interacted with the diols of the carbohydrates, as indicated by the changes in the region of carbons C2, C3 and C5, particularly in the spectrum of the F3.5 sample, corroborating the results of the thermal analysis where an additional event in the DTG curve of the sample was attributed to the crosslinks formed between PDBA and the polysaccharides of the film matrix.

#### 3.2.4. Moisture Content and Water Solubility

Data regarding moisture content (Table 5) suggest that Ca^2+^ ions indeed promoted crosslinking between the polymer chains, as they significantly reduced the moisture content in the samples compared to the control film. According to Jiang et al. [58], the decreased moisture content is an indicator that metal ions acted as crosslinking agents, while an increase suggests that such ions played a role as plasticizers in the matrix.

The control film had a higher moisture content than the other CaCl_2_-crosslinked films, and F111 had the lowest moisture content in this group. In general, the moisture content data of the samples support the results of the thermal analysis (Section 3.2.2—Figure 3), except for the moisture contents of the control film, which presented a higher content than that indicated by the TGA curve, approximately 15%.

On the other hand, CaCl_2_-crosslinked films exhibited higher water solubility than the CF sample. This result is probably due in large part to the calcium ions in the crosslinked bonds, which migrated to water during the WS test, given that the type of crosslinking using ions can be reversible [31,61]. However, the water solubility of the samples is still reduced by the three-dimensional network formed by the bonds between the polymers and calcium ions. Therefore, the stronger and better established the bonds formed by crosslinking conditions, such as the time and concentration of the bath, the lower the WS of the films will be [69].

From this, it can be inferred that the migration of calcium ions to an aqueous environment was favored in F411 since probably the percentage of calcium ions present in the sample matrix acted as a plasticizer as discussed in the FTIR and thermal analysis in Section 3.1 and Section 3.2, respectively. Furthermore, plasticizers reduce the intensity of the inter- and intramolecular bonds between polymer chains, which in turn, facilitates the entry of water molecules and consequently, favors water solubility [58].

Regarding the moisture content of PBDA-crosslinked films, there was no significant difference between CF and F2.5 (*p* > 0.05), while F3.5 and F5 showed lower moisture contents than CF, as seen in Table 1. As aforementioned, when crosslinking occurs between the polymer chains of the material, the tendency is to reduce the moisture content as occurred in F3.5 and F5. The moisture content decrease is related to the reduction in free hydroxyls in the polymer chains due to interactions with the crosslinking agent [70]. This way, this result indicates that crosslinking with PDBA was formed in the matrix of samples F3.5 and F5.

WS should decrease due to the crosslinking that forms in the matrix, as reported in the literature [30]. However, this decrease was not observed, and the WS results suggest that the water-soluble compounds may not be the same ones that interact with the crosslinking agents.

#### 3.2.5. Water Vapor Permeability (WVP)

The CaCl_2_-crosslinked films showed lower WVP values than the control film, which demonstrates that polymers crosslinked by calcium ions enhanced the water vapor barrier of the films (Table 5). However, there was no significant difference among the samples immersed in a bath of CaCl_2_ solution with different concentrations and times at a 5% significance level. The WVP reduction can be attributed to crosslinking, which reduces mobility between the macromolecule chains and due to a decrease in the void spaces in the polymer matrix and, consequently, diminishing the diffusivity of water molecules, improving the barrier properties of the films [30]. On the other hand, when the values herein obtained for WVP for the prepared films are compared to those for polyvinyl chloride (PVC), which presents a reasonably low water vapor permeability ($WVP\approx 2.6\;g\;\mathrm{mm}/m^{2}\;\mathrm{day}\;\mathrm{kPa}$), the films developed herein present an inferior water vapor barrier [77]. PBDA-crosslinked films F2.5 and F5 showed the highest WVP coefficients among all samples, and these results are attributed to the greater distances (void spaces) between the crosslinked polysaccharide chains with PDBA when compared to that promoted by the calcium ions. This greater distance results from the larger size of the inserted PDBA molecules when compared to the size of the calcium ions. This certainly contributed to the increase in permeability of water molecules through the film matrix [78].

All the samples showed a higher standard deviation value in comparison to CF, which is related to a higher dispersion of the data. This is likely associated with the homogeneous distribution of crosslinks throughout the polymeric matrix, which could have been influenced by the complexity of a matrix such as coffee waste that is comprised of various compounds such as cellulose, galactomannan and arabinogalactan polysaccharides, as well as proteins, lipids, phenolic compounds and melanoidins. Hence, the matrix of the films being heterogeneous could have interfered with the homogeneous distribution of crosslinks throughout the films. Therefore, the density of crosslinks may not be the same at all points of the sample [19].

#### 3.2.6. Stability in Acidic and Alkaline Solutions

The stability test of films under different pH conditions is relevant, considering the objective of developing biopolymer-based materials to replace traditional materials in areas where such materials will come in contact with products with a wide pH range, such as in food, pharmaceutical or cosmetic products.

CF samples did not show a clear increase in diameter during the test of stability at pH 3 and 10, and there were no statistically significant differences between the acidic or alkaline samples (*p* > 0.05). However, CF under neutral pH swelled and reached a diameter of about 20 mm. The results suggest the hydrophilic nature of the matrix and the predominant presence of neutral compounds (Table 6). In contrast, the samples crosslinked with Ca^2+^ ions did not exhibit an increase in diameter under acidic, neutral or alkaline solutions during the evaluated period. Therefore, the crosslinking provided the films with greater resistance to swelling at a neutral pH, probably by reducing the free hydroxyls in the compounds, as well as increasing the degree of compaction between the polymer chains, reducing free volume and void spaces in the film matrix [70,78,79]. The results demonstrated both the control and crosslinked films resistance to different pH values, not disintegrating during the 10-day immersion period in acidic, neutral or alkaline solutions, as can be viewed in Appendix A.

On the other hand, in general, the films crosslinked with PDBA showed greater diameters than the aforementioned samples (Table 7; Appendix A). Nevertheless, at the end of the experiment, all films remained intact and without noticeable cracks, being resistant to the tested pH values as observed in Appendix A. The films were resistant even when immersed in alkaline solutions, where the hydroxyls in the medium can destroy the hydrogen bonds and decrease the intra- and intermolecular interactions between the polymers [80].

#### 3.2.7. Morphological Analysis

Figure 7 presents the SEM images of the surface and interface of the CF films and CaCl_2_-crosslinked samples. Scanning electron microscopy (SEM) showed images of heterogeneous films with grooves and waves, which may be due to the thickness and drying rate of the samples during film formation [81]. Additionally, the reason for these imperfections can be promoted by the degree of intermolecular force between the polymer chains and the rearrangement of molecules, which can cause disordered aggregations of macromolecules during film formation [82]. Moreover, the heterogeneity of the films present in all samples may also have occurred due to the complex coffee matrix that, even after alkaline treatment, was still comprised of different polymers such as cellulose, galactomannans, arabinogalactans, as well as melanoidins, lipids, proteins and phenolic compounds. Such compounds may have had incompatibility among themselves during the formation of the polymer matrix in the film preparation process, and the polymers with more affinity may have formed aggregates as observed in the control film in Figure 7(A2).

It can be observed that the crosslinking modified the morphology of the films. CF presented a surface with grooves as seen at a magnitude of 1000× (Figure 7(A1)) and roughness at 10,000× (Figure 7(A2)), whereas the films treated with calcium chloride showed a structure with many cracks and appeared to have a thin film layer overlaying the matrix.

The images for F11 (Figure 7(B3,B4)) and, particularly, for F211 (Figure 7(C3,C4)) interfaces suggest that treatments with calcium chloride facilitated the formation of a multilayer structure, possibly induced by the complexation of the calcium ions with distinct polysaccharide chains in the matrix [83]. Xu et al. [32] also observed morphological changes in the films dissolved in a ZnCl_2_ solution after addition of CaCl_2_. The authors analyzed images with a magnitude of 40,000× and noted the formation of nanofibrils, probably related to the crosslinking formed by the ion/polymer complex.

Regarding the interface of F411 film, denser and more uniform tops and bases than of the F111 and F211 samples were observed, probably due to the Ca^2+^ ions acting not only as a crosslinking agent, but also as a plasticizer, reducing intermolecular forces, and improving the arrangement between the polymer chains. Therefore, it produced a microstructure with a more uniform and dense appearance [58,69].

In addition, CF exhibited a cross-section with a dense and polymer-rich phase, and a polymer-poor phase, where the pores of the matrix are concentrated. Both CF and F411 samples showed an interface change along the cross-section, from the denser and more compact top to the porous base. Such a structure is typical of films produced by phase inversion using the Loeb–Sourirajan technique to produce an anisotropic membrane [84]. The F2.5 film also presented this typical morphological interface (Figure 8(B3,B4)), being more similar to CF than the other PDBA-crosslinked samples. On the other hand, F3.5 and F5 showed a more uniform cross-section with smaller pores.

Regarding the surfaces, the PDBA-crosslinked samples presented a different structure from the Ca^2+^ films due to the nature of the formed crosslinks. The surfaces of F2.5 and F3.5 were irregular and rougher than the CF surface. Regarding F5, it was not possible to evaluate its matrix surface, as clusters of PDBA seem to be distributed over the entire surface of the material as seen in Figure 8(D1,D2). The higher concentration of PDBA possibly saturated the filmogenic solution, and then, a percentage of PDBA molecules probably did not participate in the three-dimensional network formation between the matrix polymers. It indicates that molecules of PDBA bonded to the surface of the material.

#### 3.2.8. Biodegradability Tests

Biodegradability tests are important to determine the potential of a material to be used as an alternative to synthetic packaging that takes a long time to degrade [85]. The biodegradability of a material can be influenced by the presence of gases, moisture content of the environment and incident light, and can be confirmed by the loss of mechanical properties, fragmentation of the material or chemical modification caused by the action of microorganisms and enzymes present in nature.

The macroscopic changes that occurred over time as the films remained buried in synthetic soil can be viewed in Appendix A. The loss of mass of the buried films over time could not be recorded, as parts of the soil clumped on the samples. Such soil particles would lead to an overestimation of the mass values, and it was not possible to remove them without damaging the film.

CF samples were monitored up to the 5th day, as it was not possible to find fragments of the specimens on the following days. To track when the film could still be found in the soil, tests were performed to unearth the CF samples on the 7th day, and then small fragments of the films were found. However, separation of the samples from the soil was not feasible due to the disintegration of the films into small particles, making the test difficult to record at this time. Therefore, it was considered that the complete biodegradation of CF occurred between the 7th and 10th day when it was not possible to find any fragments for evaluation. On the other hand, the other samples required more time to degrade, due to the crosslinking formed in the matrices.

Sample F411 showed significant signs of sample degradation from the first analysis carried out after 5 days. It was consistent with the thermal analysis results, which suggest that some of the Ca^2+^ ions acted as plasticizers and reduced the thermal stability of the F411 films. Plasticizers increase the mobility between polymer chains, favoring the diffusion of water molecules from the moist soil into the matrix, enabling microbial growth and thus accelerating the biodegradability of the samples [70].

On the other hand, fragments of the F211 sample were still found after 6 months. This result must be due to the crosslinks formed in the film matrix corroborating the analysis of the NMR spectra (Figure 6), which show F211 as the sample with the greatest compatibility among compounds due to the crosslinking promoted by the calcium chloride bath. Furthermore, the thermal analysis results are consistent with the biodegradability test since F211 showed greater thermal stability and was more resistant to degradation.

The F5 film took longer to fully degrade, and fragments of the test could still be found even after 180 days of the sample being buried in the soil. Except for F211 and F5, the others were not found in the last evaluation proposed in the methodology, suggesting that these samples were probably fully degraded between 3 and 6 months.

Samples F211 and F5 related to the treatment with 10% CaCl2 (*m*/*v*) for 5 min and to the treatment with 5% 1,4-phenilenediboronic acid solution, respectively, were not strictly considered biodegradable according to European Norm EN13432. According to the European standard EN13432 for biodegradability requirements, the material is considered biodegradable if at least 90% of its mass (organic carbon-based content) is decomposed within 6 months (180 days) [86]. Therefore, F211 was deemed non-biodegradable within this period (180 days) and this was attributed to the fact that, due to the short time of exposure to calcium ions, the polysaccharide molecules remained more tightly packed in the film structure compared to the other films in lieu of less calcium ions crosslinking the molecule chains, thus making it difficult for the degrading microorganisms to access the polysaccharide molecules. However, the F411 sample was also treated with a 10% CaCl_2_ (*m*/*v*) solution for a period of 30 min, thus allowing for more inclusions of calcium ions between the polysaccharides’ chains, increasing their distance and thus providing a pathway for water molecules in between the molecules, which further provided the degrading microorganisms with easier access to the polysaccharide molecules. F411 was deemed biodegradable according to the European Norm EN13432. Hence, it was concluded that the time length of treatment with 10% CaCl_2_ (*m*/*v*) solution was a major factor in turning the films biodegradable during a period of 180 days buried in the soil. It is noteworthy to mention that the F211 sample (and all the others as well) is in actuality biodegradable, for its degradation by microorganisms necessarily occurs in a significant shorter time than that needed for degradation of petroleum-based polymeric films presenting catatonic degradation behaviors.

Regarding sample F5, it was observed by scanning electron microscopy that the 5% 1,4-phenilenediboronic acid saturated the aqueous solution, and clusters of precipitated 1,4-phenilenediboronic acid were deposited on the surface of the film. Boronic acids are known to present antimicrobial activity and this factor was deemed responsible for extending the degradation period of sample F5 beyond 180 days. The boronic groups effectively crosslinking the polysaccharide chains were not available to act as antimicrobial agents but the deposited clusters of phenylenediboronic acid on the film surface were free to act as such.

Comparing the crosslinked films with CF, it can be inferred that crosslinking occurred in all films to some degree, as all samples degraded over a longer period than the control film. This result is consistent with the reports in the literature that crosslinking decreases the biodegradability of the material since it hinders microbial action on the solid matrix [30,39].

Based on the European standard EN13432 for biodegradability requirements, it can be stated that all films developed in this study can be considered biodegradable, except for F211 and F5. Therefore, the materials considered biodegradable can be readily discarded without industrial intervention [39]. Nonetheless, F211 and F5 can be actually considered biodegradable when their degradation period is compared to those of petroleum-derived polymers. 

#### 3.2.9. Mechanical Properties

Tensile tests were performed on the samples that were considered biodegradable, therefore, F211 and F5 were excluded from the trials. Furthermore, regarding the PDBA films, samples that did not show pronounced crosslinking formation, as suggested by the thermal analysis, were excluded. Hence, only the F3.5 sample was considered. The mechanical properties were then analyzed for samples CF, F111, F411 and F3.5 (Table 8).

Sample F111 showed the highest tensile strength (TS) among all the crosslinked films and did not present a significant difference from the control film. Furthermore, the results indicate that a lower concentration of calcium chloride and a shorter immersion time in the crosslinking solution seems to be more appropriate for increasing the tensile strength of the films. Notice that F411 was immersed in a 10% (*w*/*v*) calcium chloride solution for 30 minutes, whereas F111 was immersed in a 6% (*w*/*v*) solution for 17.5 min.

Generally, the addition of large amounts of crosslinking agents increases TS and decreases the elongation at break (EB) of films. However, in this study, the treatment with the highest concentration of the agent and the longest time did not have this effect [4,5,8]. As mentioned earlier, the calcium ions acted as plasticizers instead of crosslinking agents. In general, plasticizers decrease tensile strength and increase the elongation at break. A study carried out by Xu et al. [32], in which the methodology employed for the preparation of cellulose films was similar to the one herein employed, resulted in TS values of 0.217 to 0.882 MPa, for the control and the calcium-laden films, respectively. The inclusion of calcium ions in the structure promoted an increase of 350% in the tensile strength of the films. Research carried out on films of polysaccharides from spent coffee grounds, employing zinc chloride aqueous solution as the solvent, reported a TS value of 0.3 MPa [29]. The tensile strength of conventional petroleum-derived low-density polyethylene and polyvinyl chloride films used for food packaging are in the range of 10 to 55 MPa [87]. Scientifically speaking, films prepared from distinct precursor materials and distinct processes should not be compared amongst themselves. Nevertheless, it is plausible to do so when we talk about the applicability of films based on a single property, such as tensile strength, i.e., based solely on tensile strength, the films herein produced could be used for the same types of applications to which the conventional petroleum-derived films are applied. However, when all the different properties needed for defining a suitable application are considered, such a comparison cannot be straightforwardly made. 

Samples F3.5 and CF showed the highest moisture and humidity levels among the samples subjected to mechanical testing, indicating that both samples showed typical behavior of hydrophilic matrices. Moisture content directly affects the mechanical properties of the films, especially films of polysaccharides and proteins, as the water molecules act as a plasticizer in the matrix, reducing molecular interaction and increasing the mobility of the polymer chains, making the films more flexible [82,88].

## 4. Conclusions

The alkaline hydrogen peroxide treatment yielded 56% mass regarding the original mass of the spent coffee grounds and was shown to be effective in removing phenolics compounds and concentrate polysaccharides. The crosslinking performed by calcium ions in the biopolymer films obtained from delignified spent coffee grounds, increased the water vapor barrier properties in all treatments. Overall, it was observed that calcium ions can improve film compatibility by reducing intermolecular and intramolecular forces between the polymeric chains and promoting crosslinking. The film immersed in a 6% calcium chloride solution for 17.5 min (F111) showed the same level of tensile strength as the control film. The results suggest that Ca^2+^ ions produced a plasticizing effect on the films immersed in a 10% CaCl_2_ solution for 30 min. Regarding films developed with 1,4-phenylenediboronic acid, the results suggest that 3.5 and 5% of the reagent promoted crosslinking between polymer chains, and F3.5 showed a higher percentage of elongation than the calcium chloride films. All films developed in this study were considered biodegradable according to European standard EN13432, except for the film immersed in a 10% CaCl_2_ solution for 5 min (F211) and film with 5% of PDBA (F5). This study concludes that both films with different types of crosslinking presented advantages and disadvantages that will depend on the material application. However, further studies can be carried out to improve the developed biopolymer films for general use in the future.

## Figures and Tables

**Figure 1 foods-12-02520-f001:**
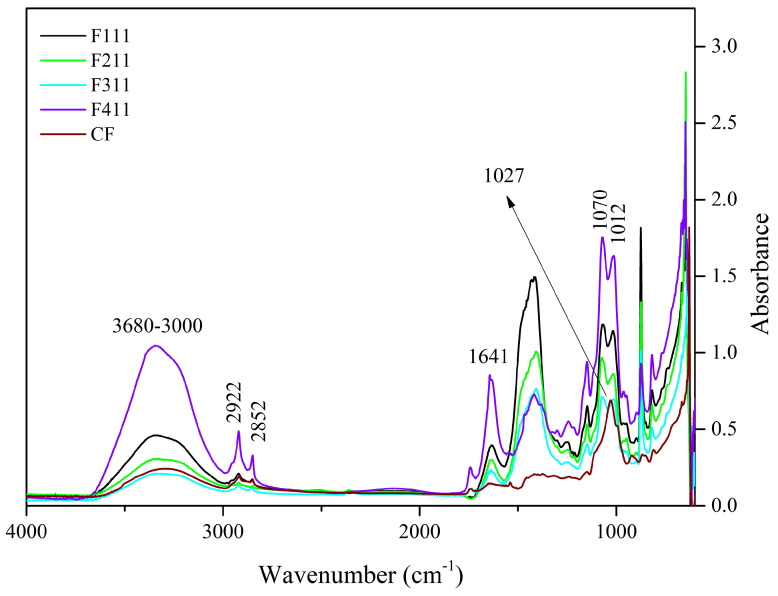
Spectra of the control film (CF) and samples crosslinked by calcium ions: F111—immersed in 6% CaCl_2_ solution for 17.5 min; F211—immersed in 10% CaCl_2_ solution for 5 min; F311—immersed in 2% CaCl_2_ solution for 30 min; F411—immersed in 10% CaCl_2_ solution for 30 min.

**Figure 2 foods-12-02520-f002:**
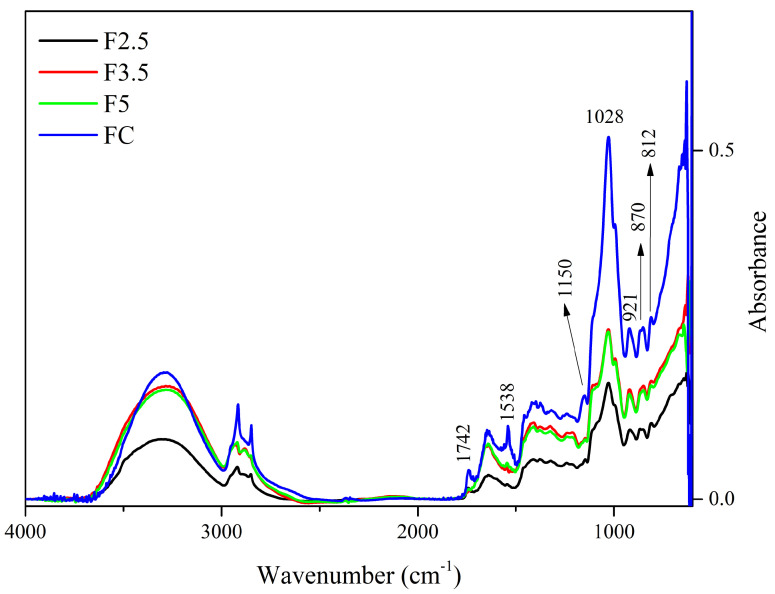
Spectra of the control film (CF) and samples crosslinked by 1,4-phenilenediboronic acid (PDBA): F2.5—addition of 2.5% PDBA; F3.5—addition of 3.5% PDBA; F5—addition of 5.0% PDBA.

**Figure 3 foods-12-02520-f003:**
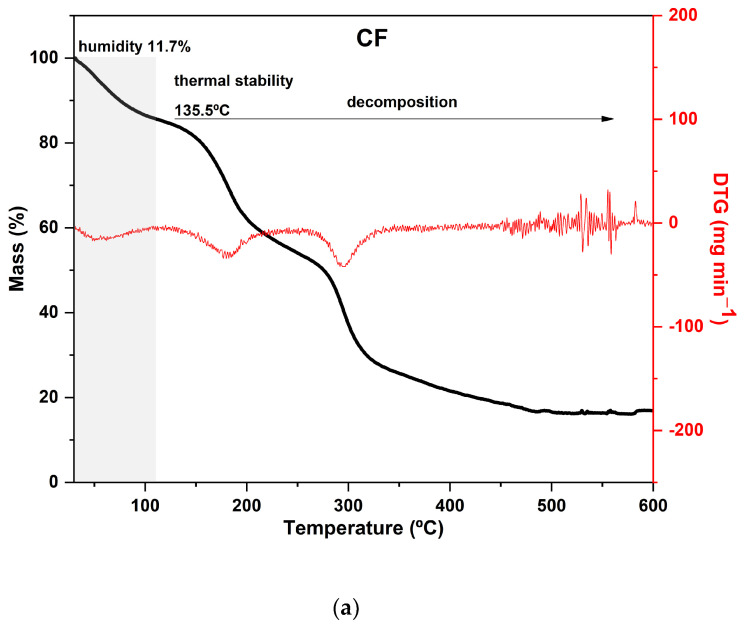
Curves of thermogravimetric analysis (black) and (red) derivative thermogravimetry (DTG) of the following samples: (**a**) control film (CF) and samples crosslinked by calcium ions: (**b**) F111—immersed in 6% CaCl_2_ solution for 17.5 min; (**c**) F211—immersed in 10% CaCl_2_ solution for 5 min; (**d**) F311—immersed in 2% CaCl_2_ solution for 30 min; (**e**) F411—immersed in 10% CaCl_2_ solution for 30 min; (**f**) F511—immersed in 2% CaCl_2_ solution for 5 min.

**Figure 4 foods-12-02520-f004:**
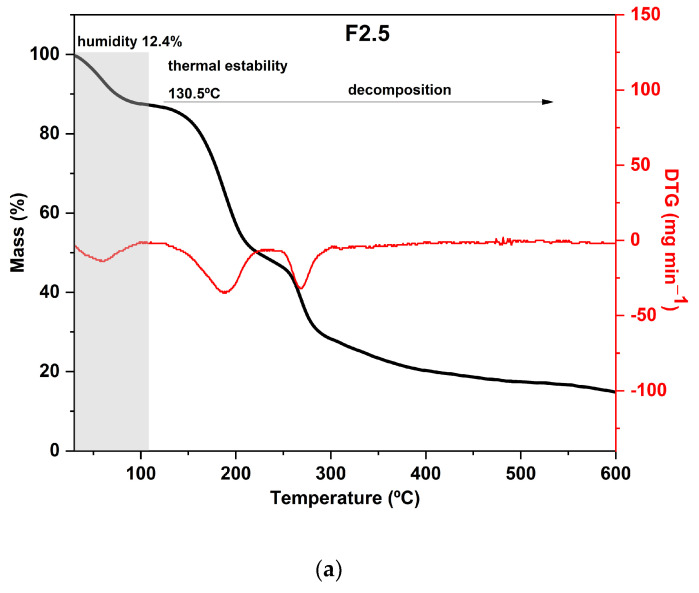
Curves of thermogravimetric analysis (black) and (red) derivative thermogravimetry (DTG) of the following samples: (**a**) F2.5—addition of 2.5% PDBA; (**b**) F3.5—addition of 3.5% PDBA; (**c**) F5—addition of 5.0% PDBA.

**Figure 5 foods-12-02520-f005:**
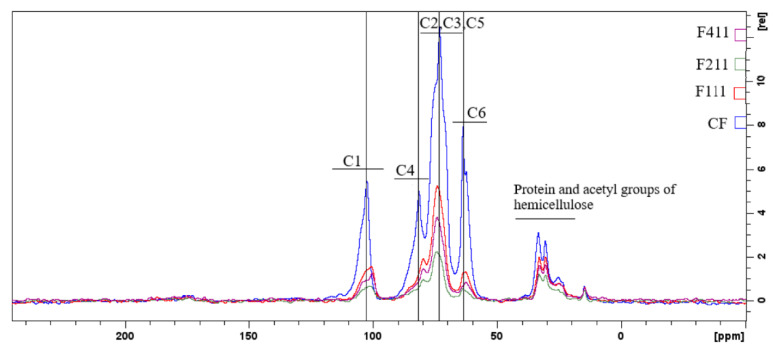
Spectra of state solid RMN 13C of the following samples: control film (CF) and samples crosslinked by calcium ions: F111—immersed in 6% CaCl_2_ solution for 17.5 min; F211—immersed in 10% CaCl_2_ solution for 5 min; F411—immersed in 10% CaCl_2_ solution for 30 min.

**Figure 6 foods-12-02520-f006:**
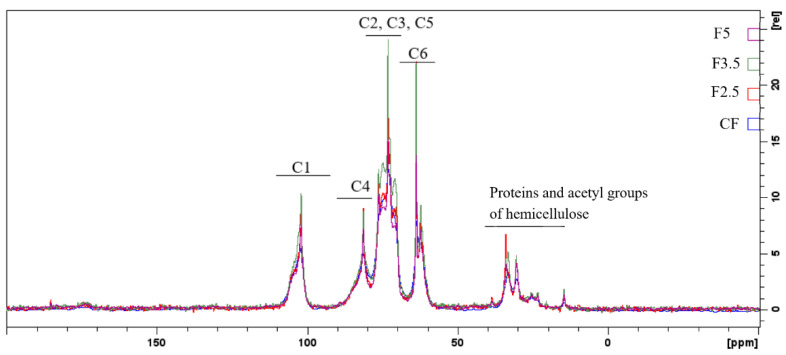
Spectra of state solid RMN 13C of the following samples: CF—control film and samples crosslinked by 1,4-phenilenediboronic acid (PDBA): F2.5—addition of 2.5% PDBA; F3.5—addition of 3.5% PDBA; F5—addition of 5.0% PDBA.

**Figure 7 foods-12-02520-f007:**
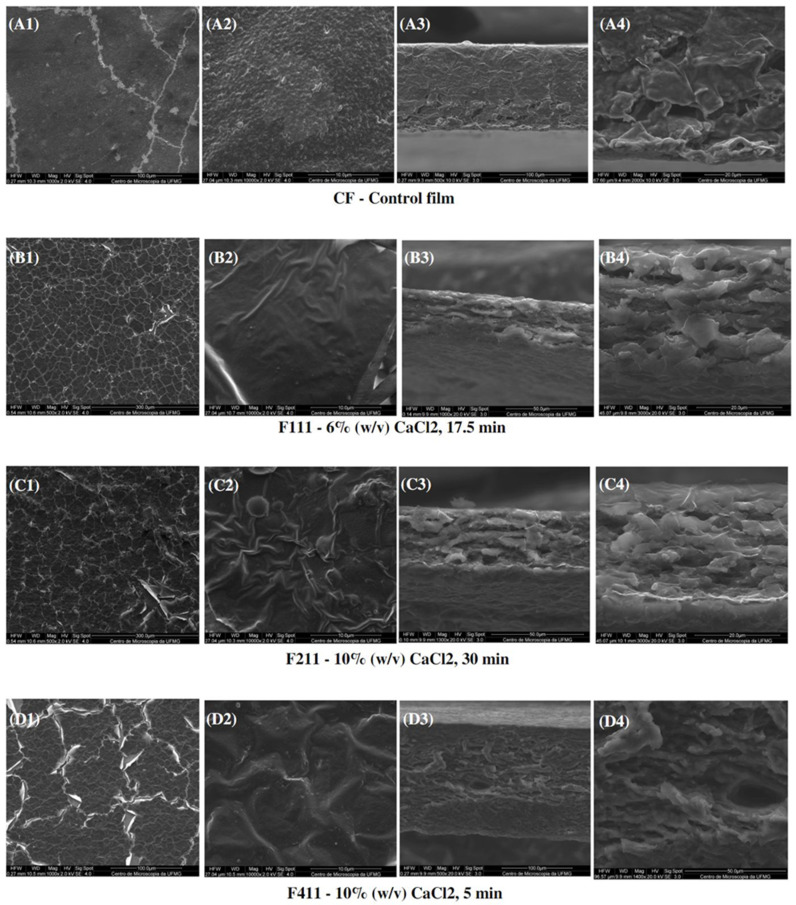
SEM micrographs of surfaces (1,2) and cross-sections (3,4) of the control film (CF) and CaCl_2_-crosslinked films.

**Figure 8 foods-12-02520-f008:**
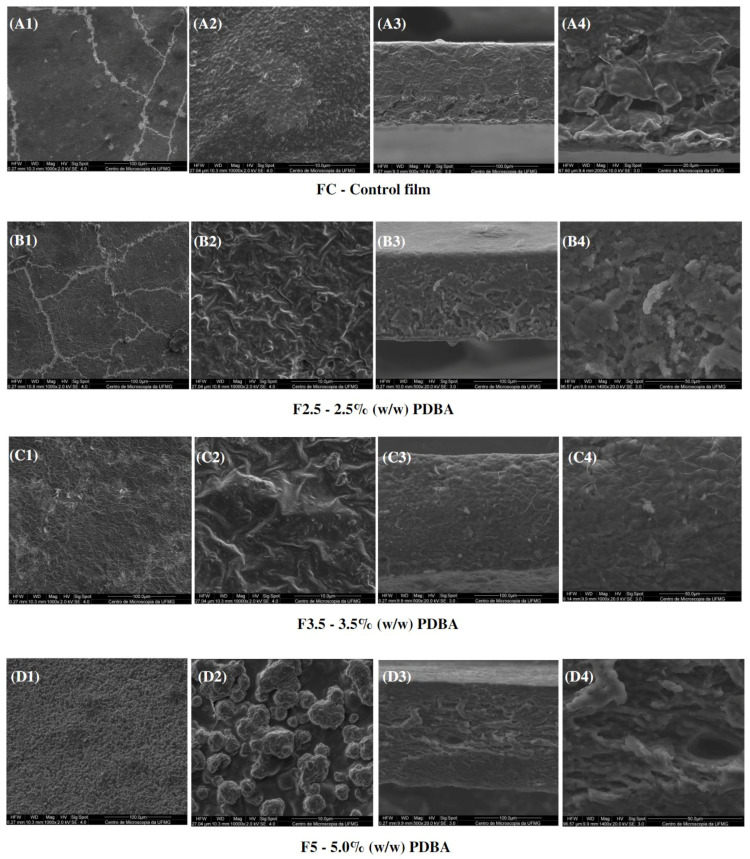
SEM micrographs of s surfaces (1,2) and cross-sections (3,4) of the control film and PDBA-crosslinked films.

**Table 1 foods-12-02520-t001:** Samples submitted to different concentrations and times of an immersion bath of CaCl_2_ solution.

Samples	Concentration (% *w*/*v*)	Time (min)
F111	6	17.5
F211	10	5
F311	2	30
F411	10	30
F511	2	5

**Table 2 foods-12-02520-t002:** Samples crosslinked to different concentrations of 1,4-phenilenediboronic acid (PBDA).

Samples	Concentration of PBDA (% *w*/*w* of SCGs-AHP)
F2.5	2.5
F3.5	3.5
F5	5.0

**Table 3 foods-12-02520-t003:** Composition of spent coffee grounds (SCGs) and spent coffee grounds submitted to alkaline hydrogen peroxide treatment (SCGs-AHP).

Sample	Composition (% *w*/*w*)					
	Carbohydrate	Protein	Lipids	Phenolics	Moisture	Ash
SCGs	46.31	13.00 ± 0.50 ^a^	13.77 ± 0.42 ^a^	21.42 ± 2.39	6.52 ± 0.03 ^b^	1.84 ± 0.10 ^a^
SCGs-AHP	63.41	3.43 ± 0.19 ^b^	14.68 ± 1.22 ^a^	9.15 ± 1.01 ^b^	7.51 ± 0.08 ^a^	2.08 ± 0.06 ^a^

Mean ± standard deviation (*n* = 3). Different letters in the same column indicate that values are significantly different (*p* > 0.05) by Student’s *t*-test paired.

**Table 4 foods-12-02520-t004:** Sugar analysis of spent coffee grounds (SCGs) and spent coffee grounds submitted to alkaline hydrogen peroxide treatment (SCGs-AHP).

Sample	Monosaccharide Composition (% mol)				
	Glucose	Mannose	Galactose	Arabinose	Total
SCGs	25.68 ± 0.32 ^b^	37.90 ± 0.20 ^a^	17.63 ± 0.09 ^b^	18.78 ± 0.17 ^a^	100
SCGs-AHP	31.41 ± 0.85 ^a^	36.35 ± 0.53 ^b^	19.43 ± 0.28 ^a^	12.81 ± 0.19 ^b^	100

Mean ± standard deviation (*n* = 3). Different letters in the same column indicate that values are significantly different (*p* > 0.05) by Student’s *t*-test paired.

**Table 5 foods-12-02520-t005:** Moisture content and water solubility (WS) of the control film (CF), films immersed in a chloride calcium solution (F111—6%, 17.5 min; F211—10%, 5 min; F311—2%, 30 min; F411—10%, 30 min; F511—2% 5 min), and samples with 1,4-phenilenediboronic acid (PDBA) (F2.5—2.5% PDBA; F3.5—3.5% PDBA; F5—5.0% PDBA).

Sample	Moisture Content (%)	Water Solubility (%)	WVP (g mm/m^2^ Day kPa)
CF	42.60 ± 0.52 ^aA^	43.95 ± 3.16 ^cA^	45.32 ± 3.77 ^cB^
F111	20.30 ± 0.11 ^c^	52.48 ± 1.40 ^b^	52.48 ± 1.40 ^b^
F211	27.83 ± 2.34 ^b^	50.91 ± 3.67 ^bc^	50.91 ± 3.67 ^bc^
F411	25.23 ± 1.88 ^b^	61.00 ± 1.94 ^a^	61.00 ± 1.94 ^a^
F2.5	49.01 ± 2.54 ^A^	48.71 ± 8.65 ^A^	53.26 ± 3.54 ^A^
F3.5	30.71 ± 3.78 ^B^	39.37 ± 2.90 ^A^	39.37 ± 2.90 ^B^
F5	30.75 ± 4.26 ^B^	51.68 ± 4.02 ^A^	51.68 ± 4.02 ^A^

Results are expressed as mean ± standard deviation. The lowercase letters in the columns correspond to the samples treated with calcium chloride, while the uppercase letters correspond to the samples cross-linked via 1,4-phenilenediboronic acid. Different letters in the same column indicate statistically significant differences by Tukey Test (*p* < 0.05).

**Table 6 foods-12-02520-t006:** The results of the stability test of control film (CF) and CaCl_2_-crosslinked samples (F111—6%, 17.5 min; F211—10%, 5 min; F311—2%, 30 min; F411—10%, 30 min; F511—2% 5 min) after 10 days immersed under different pH values.

Sample	Diameter (mm)		
	pH 3	pH 7	pH 10
CF	16.6 ± 0.2 ^Ba^	20.2 ± 0.6 ^Aa^	17.3 ± 0.2 ^Ba^
F111	16.2 ± 0.1 ^Aa^	16.1 ± 0.2 ^Ab^	16.5 ± 0.4 ^Ab^
F211	16.3 ± 0.3 ^Aa^	16.4 ± 0.2 ^Ab^	16.0 ± 0.3 ^Ab^
F411	16.5 ± 0.4 ^Aa^	16.5 ± 0.5 ^Ab^	16.2 ± 0.1 ^Ab^

Results are expressed as mean ± standard deviation. Different lowercase letters in the same column indicate statistically significant differences by Tukey Test (*p* < 0.05). Different uppercase letters in the same line indicate statistically significant differences by Tukey Test (*p* < 0.05).

**Table 7 foods-12-02520-t007:** The results of the stability test of the control film (CF) and the films crosslinked by 1,4-phenylenediboronic acid (PBDA) (F2.5—2.5% PDBA; F3.5—3.5% PDBA; F5—5.0% PDBA) after 10 days immersed under different pH.

Sample	Diameter (mm)		
	pH 3	pH 7	pH 10
CF	16.6 ± 0.2 ^Ba^	20.2 ± 0.6 ^Aa^	17.3 ± 0.2 ^Bb^
F2.5	19.3 ± 1.5 ^ABa^	17.3 ± 0.5 ^Bb^	20.6 ± 0.8 ^Aa^
F3.5	18.9 ± 1.8 ^ABa^	16.5 ± 0.2 ^Bb^	20.1 ± 0.6 ^Aa^
F5	18.8 ± 0.2 ^Aa^	19.9 ± 1.3 ^Aa^	19.1 ± 0.5 ^Aa^

Results are expressed as mean ± standard deviation. Different lowercase letters in the same column indicate statistically significant differences by Tukey Test (*p* < 0.05). Different uppercase letters in the same line indicate statistically significant differences by Tukey Test (*p* < 0.05).

**Table 8 foods-12-02520-t008:** Mechanical properties of the control film (CF), films immersed in a chloride calcium solution (F111—6%, 17.5 min; F411—10%, 30 min) and samples with 1,4-phenilenediboronic acid (PDBA) (F3.5—3.5% PDBA).

Sample	Thickness (mm)	Tensile Strength (MPa)	Elongation at Break (%)
CF	0.215 ± 0.067	2.448 ± 0.387 ^ab^	48.067 ± 10.399 ^a^
F111	0.178 ± 0.010	3.087 ± 0.418 ^a^	15.182 ± 1.367 ^bc^
F411	0.228 ± 0.019	2.041 ± 0.266 ^b^	9.790 ± 1.160 ^c^
F3.5	0.231 ± 0.035	1.341 ± 0.281 ^c^	25.527 ± 2.587 ^b^

Results are expressed as mean ± standard deviation (*n* = 5). Different letters in the same column indicate statistically significant differences by Tukey Test (*p* < 0.05).

## Data Availability

Data is contained within the article or Appendix A.

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
