# Peer review of "Development of Films from Spent Coffee Grounds’ Polysaccharides Crosslinked with Calcium Ions and 1,4-Phenylenediboronic Acid: A Comparative Analysis of Film Properties and Biodegradability"

_foods, 2023, doi:10.3390/foods12132520_

Round 1

Reviewer 1 Report

The objectives are clear and well defined, in this work was to d to develop films from spent coffee grounds (SCG) polysaccharide fraction, which is comprised of cellulose, galactomannans and arabinogalactans.

The manuscript is original, and represents a good contribution, addition of knowledge to scientific literature of biodegradable. Compared g films developed with 1,4-phenylenediboronic acid, results suggest that of the reagent promoted crosslinking between polymer chains, showed a higher percentage of elongation than the calcium chloride films.

In addition, it would be necessary to clarify some points

1) In general, the TGA analysis provides relatively little information, once the decomposition has started and it is difficult to interpret. The TGA is an interesting tool to know the humidity, but it loses interest once the decomposition starts. For example, as the residual in figure 3 is interpreted, the CF has a residual of 20%, the F511 50%. For what reason? It would have been interesting to complete it with DSC

2) Why are some materials biodegradable and others not?

The strengths of the method described in the manuscript in the experimental results obtained and correct methodology. On another side, the main problem with the work is that it is very specific. This means that outside certain areas of study the interest may be low. About the change and required information, as follow: More information should be included about material behavior

Author Response

The objectives are clear and well defined, in this work was to d to develop films from spent coffee grounds (SCG) polysaccharide fraction, which is comprised of cellulose, galactomannans and arabinogalactans.

The manuscript is original, and represents a good contribution, addition of knowledge to scientific literature of biodegradable. Compared g films developed with 1,4-phenylenediboronic acid, results suggest that of the reagent promoted crosslinking between polymer chains, showed a higher percentage of elongation than the calcium chloride films.

In addition, it would be necessary to clarify some points:

1) In general, the TGA analysis provides relatively little information, once the decomposition has started and it is difficult to interpret. The TGA is an interesting tool to know the humidity, but it loses interest once the decomposition starts. For example, as the residual in figure 3 is interpreted, the CF has a residual of 20%, the F511 50%. For what reason? It would have been interesting to complete it with DSC.

The difference in the amount of residue between CF and F511 is attributed to the higher thermal stability of the F511 film due to its higher density of crosslinks promoted by the calcium ions, proving the efficacy of the proposed treatments. This information was inserted in the text and the original figures were replaced by modified ones for a better understanding of the films’ thermal behavior.

In previous works by our research group [1,2], we have demonstrated the films produced from polysaccharide-rich fraction of spent coffee grounds to be completely amorphous, meaning that, in the process of film production by precipitation of a casting solution in a nonsolvent bath, there was no sufficient time for the dissolved constituent polysaccharides (i.e., cellulose and galactomannan) to recrystallize into their original crystalline forms or into any other alternative crystalline form (e.g., from original cellulose I to cellulose II crystalline form). This fact was attributed to the higher solution viscosity of the dissolved polysaccharides, which limits molecular dynamics and thus preclude the molecules to kinetically achieve the most energetically favorable state during the precipitation of the casting solution and subsequent film forming step. Also, it should be duly noted that there was a significant amount of zinc ions embedded in the polymeric network that most certainly hindered the crystallization of the dissolved polysaccharide chains. Although it was expected that a complete amorphous polymer network would allow for an easy determination of its glass transition temperature (or temperatures for that matter, given the complexity of the matrix), our attempts to do so in our current work were unsuccessful, with the results rendering inconclusive. DSC analyses of the films produced from spent coffee ground polysaccharides in our previous works [1,2] have demonstrated their glass transition temperatures to be in the range of  to . Therefore, the obtained TG information was thus herein deemed sufficient to describe the thermal behavior of the films, correctly reflecting the distinct changes in molecular structure caused by distinct crosslinking reaction conditions, including the use of distinct crosslinking agents.

  1. Batista, M.J.P.A.; Ávila, A.F.; Franca, A.S.; Oliveira, L.S. Polysaccharide-Rich Fraction of Spent Coffee Grounds as Promising Biomaterial for Films Fabrication. Carbohydr. Polym. 2020, 233, 115851, doi:10.1016/j.carbpol.2020.115851.
  2. Coelho, G.O.; Batista, M.J.A.; Ávila, A.F.; Franca, A.S.; Oliveira, L.S. Development and Characterization of Biopolymeric Films of Galactomannans Recovered from Spent Coffee Grounds. J. Food Eng. 2021, 289, 110083, doi:10.1016/j.jfoodeng.2020.110083.

2) Why are some materials biodegradable and others not?

Samples F211 and F5, respectively related to the treatment with 10 % CaCl2 (m/v) for 5 min and to the treatment with 5 % 1,4-phenilenediboronic acid solution, were not considered biodegradable strictly according to European Norm EN13432. According to the European standard EN 13432 for biodegradability requirements, the material is considered biodegradable if at least 90 % of its mass (organic carbon-based content) is decomposed within 6 months (180 days). Therefore, F211 was deemed non-biodegradable within this period (180 days) and this was attributed to the fact that, due to the short time of exposure to calcium ions, the polysaccharide molecules remained more tightly packed in the film structure compared to the other films in lieu of less calcium ions crosslinking the molecule chains, thus making it difficult for the degrading microorganisms to access the polysaccharide molecules. F411 sample was also treated with a 10 % CaCl2 (m/v) solution, however, for a period of 30 min, thus allowing for more inclusions of calcium ions between the polysaccharides’ chains, increasing their distance and thus providing a pathway for water molecules in between the molecules, which further provided the degrading microorganisms easier access to the polysaccharide molecules. F411 was deemed biodegradable according to the European Norm EN13432. Hence, it was concluded that the time length of treatment with 10 % CaCl2 (m/v) solution was a major factor in turning the films biodegradable during a period of 180 days buried in the soil. It is noteworthy to mention that F211 sample (and all the others as well) is in actuality biodegradable, for its degradation by microorganisms necessarily occurs in a significant shorter time than that needed for degradation of petroleum-based polymeric films presenting catatonic degradation behaviors.

Regarding sample F5, it was observed by scanning electron microscopy that the 5 % 1,4-phenilenediboronic acid saturated the aqueous solution, and clusters of precipitated 1,4-phenilenediboronic acid were deposited on the surface of the film. Boronic acids are known to present antimicrobial activity and this factor was deemed responsible for extending the degradation period of sample F5 beyond 180 days. The boronic groups effectively crosslinking the polysaccharide chains were not available for acting as antimicrobial agents but the deposited clusters of phenilenediboronic acid on the film surface were free to act as such [1-3].

  1. Sayin, Z.; Ucan, U.S.; Sakmanoglu, A. Antibacterial and Antibiofilm Effects of Boron on Different Bacteria. Biol. Trace Elem. Res. 2016, 173, 241–246, doi:10.1007/s12011-016-0637-z.
  2. Defrancesco, H.; Dudley, J.; Coca, A. Boron Chemistry: An Overview. ACS Symp. Ser. 2016, 1236, 1–25, doi:10.1021/bk-2016-1236.ch001.
  3. Halbus, A.F.; Horozov, T.S.; Paunov, V.N. Strongly Enhanced Antibacterial Action of Copper Oxide Nanoparticles with Boronic Acid Surface Functionality. ACS Appl. Mater. Interfaces 2019, 11, 12232–12243, doi:10.1021/acsami.8b21862.

The strengths of the method described in the manuscript in the experimental results obtained and correct methodology. On another side, the main problem with the work is that it is very specific. This means that outside certain areas of study the interest may be low. About the change and required information, as follow: More information should be included about material behavior.

We have failed to understand what the Reviewer meant by “the main problem with the work is that it is very specific”, so we kindly ask the Reviewer to clarify what is the rationale behind this statement, in a way that the issue can be properly addressed. The Authors are open to scientific discussion and believe that more elaborate and detailed comments from the Reviewers can significantly improve the manuscript. What kind of information about material behavior is required by the Reviewer to be further discussed in the manuscript?

From a scientific standpoint, our work has contributed to further the knowledge in the general areas of biopolymeric films and valorization/upcycling of agri-food co-products and wastes. The research work presented in the manuscript is completely within the scope of the special issue, which includes processing of coffee wastes (or co-products) for their further valorization and consequent cascade extension of coffee lifecycle service. Thus, we have failed to understand how the statement “outside certain areas of study the interest may be low” holds regarding our work.

Reviewer 2 Report

The paper presents a comparative study of biopolymeric films based on SCG. It is a topic of interest in researchers in the related areas, but the paper needs improvement. My detailed comments are as follows:

1. The introduction section should be rewritten extensively, describe in detail the advantages of the material, and reduce the length of irrelevant content. The introduction should end with a formulated research objective resulting from the literature review, and not with a summary of the methods used.

2. Why not use the corresponding method to prove the extracted substance?

3. Why is there only one table in the Materials and Methods section for one type of membrane?

4. There are too many tables in the text, we suggest changing the format to pictures so that people can visually see the change in trend.

5. Why are not all membranes characterized at a later stage?

Moderate editing of the English language

Author Response

The paper presents a comparative study of biopolymeric films based on SCG. It is a topic of interest in researchers in the related areas, but the paper needs improvement. My detailed comments are as follows:

  1. The introduction section should be rewritten extensively, describe in detail the advantages of the material, and reduce the length of irrelevant content. The introduction should end with a formulated research objective resulting from the literature review, and not with a summary of the methods used.

Modified as requested. The introduction was extensively rewritten to comply with the Reviewer’s request.

  1. Why not use the corresponding method to prove the extracted substance?

We first omitted the detailed description of the spent coffee grounds polysaccharide-rich fraction extraction methodology to avoid significant inflation of the manuscript, since the employed methodology was described in detail in our previous works. However, to comply with the reviewer request, the detailed description of the referred methodology was included in the manuscript and is entitled ‘Removal of phenolics from spent coffee grounds’.

  1. Why is there only one table in the Materials and Methods section for one type of membrane?

To comply with the reviewer observation, we have included a table with treatment conditions for 1,4-phenilenediboronic acid in the associated methodology section.

  1. There are too many tables in the text, we suggest changing the format to pictures so that people can visually see the change in trend.

The majority of the Tables presented in our manuscript do not contain data that can be expressed as behavior in a plotted curve but rather present individual values of a property for distinct samples of films. The samples related to a single type of crosslinking agents, submitted to incremental values of a single treatment condition, did not allow for a construction of a continuous evolutive behavior, thus we have deemed appropriate to present the results as discreet values of a property instead of using tendency lines or bar plots. Furthermore, presenting these data in a Table allows for statistically differentiating the samples according to their respective individual properties.

  1. Why are not all membranes characterized at a later stage?

Samples F311 and F511 were rather brittle making it difficult to handle in a way that maintained their integrity throughout all the analytical procedures. Hence, these samples were discarded as not useful for the intended future applications for the prepared films. The experiments for determination of mechanical properties were used as criterion for decision making about further using or not a specific sample to carry out all the other properties’ measurements. Hence, brittle samples were readily discarded for they did not present desirable traits of elongation. Also, the samples previously determined as non-biodegradable (in accordance with European Norm EN13432) were not subjected to the mechanical properties test, since film biodegradability was prioritized as one of the most desired properties in our work. Therefore, samples F211 and F5 were not further tested for other properties, such as tensile strength. Regarding the films prepared with 1,4-phenilenediboronic acid as crosslinking agent, samples deemed not sufficiently crosslinked by the thermogravimetric analysis were discarded, thus, only sample F3.5 was considered for further testing.

Reviewer 3 Report

The authors studied the performance of biopolymeric film produced using spent coffee grounds. They observed that mechanical properties of the film could be enhanced using cross-linkers. However, there are disadvantages when using cross-linkers, which could lead to higher moisture absorption. 

Overall, the study is complete and useful. I suggest that the authors revise the paper based on the following comments before I can recommend the paper to be considered for the journal. 

Comment #1:

In the abstract, please include the percentages of improvement/deterioration of tested properties accordingly. The percentages would be able to reflect better the significance of the reported work.

Comment #2:

The current introduction lacks the following elements:

-In the introduction, please elaborate, specifically, what are the market needs when it comes to the performance of packaging purposes. 

-Also, do highlight and elaborate further on the specific challenges in using biopolymeric films compared to conventional ones.

Comment #3:

Typically, moisture attack on materials absorbs at different rates.  For moisture content measurement, the authors explained that the samples were left submerged for 10 days based on reference [22]. Is the moisture content saturation achieved within this duration? Can the authors clarify?

Comment #4:

Following on from the previous comment, did the authors tested the samples with or without moisture content? There is confusion because the authors explained that samples F3.5 and CF showed higher moisture contents, indicating the hydrophilic nature that affects the samples’ mechanical properties. If the authors did the mechanical testing using new samples (not those which undergone moisture tests), then I would suggest that the authors provide the moisture content information of the samples to be exact.

Comment #5:

How does the properties of the biopolymeric film compare with conventional ones? I suggest that the authors discuss the comparison. This will allow for a better judgement as to how significant is the improvement obtained from the present study.

Moderate english editing will help improve the paper.

Author Response

The authors studied the performance of biopolymeric film produced using spent coffee grounds. They observed that mechanical properties of the film could be enhanced using cross-linkers. However, there are disadvantages when using cross-linkers, which could lead to higher moisture absorption.

Overall, the study is complete and useful. I suggest that the authors revise the paper based on the following comments before I can recommend the paper to be considered for publication with the journal.

Comment #1:

In the abstract, please include the percentages of improvement/deterioration of tested properties accordingly. The percentages would be able to reflect better the significance of the reported work.

Modified as requested.

Comment #2:

The current introduction lacks the following elements:

-In the introduction, please elaborate, specifically, what are the market needs when it comes to the performance of packaging purposes.

-Also, do highlight and elaborate further on the specific challenges in using biopolymeric films compared to conventional ones.

Modified as requested.

Comment #3:

Typically, moisture attack on materials absorbs at different rates.  For moisture content measurement, the authors explained that the samples were left submerged for 10 days based on reference [22]. Is the moisture content saturation achieved within this duration? Can the authors clarify?

The reviewer might be mixing up two different methodologies employed for two different properties. The moisture contents of the prepared films were determined by a conventional convective oven drying method. Due to differences in composition of the films, they necessarily will present distinct initial moisture contents and distinct equilibrium moisture contents for the exact same drying conditions. The films were submerged in acidic, neutral and alkaline solutions for 10 days for the determination of their stability in such solutions. There is no link between the methodology for determination of moisture content and the methodology for determination of film stability in aqueous solutions.

Comment #4:

Following on from the previous comment, did the authors tested the samples with or without moisture content? There is confusion because the authors explained that samples F3.5 and CF showed higher moisture contents, indicating the hydrophilic nature that affects the samples’ mechanical properties. If the authors did the mechanical testing using new samples (not those which undergone moisture tests), then I would suggest that the authors provide the moisture content information of the samples to be exact.

The following information was added to section 2.4.10 Mechanical Properties to clarify such issue: “Films were tested with their moisture contents as determined by the gravimetric pro-cedure described in section 2.4.4.”

The tests were performed at different moisture contents because each individual moisture content is the equilibrium moisture content of the respective film. Recall that all the films were equilibrated at the exact same temperature and relative moisture content of the air, for the same period. Thus, even though moisture content is known to affect the film glass transition temperature and consequently all the other properties, it does not make sense to match the moisture contents of the films if for any given set of atmospheric conditions the films’ equilibrium moisture content will be necessarily different from each other.

Comment #5:

How does the properties of the biopolymeric film compare with conventional ones? I suggest that the authors discuss the comparison. This will allow for a better judgement as to how significant is the improvement obtained from the present study.

Modified as requested. The following text was added to the manuscript: “A study carried out by Xu et al. [32], in which the methodology employed for the preparation of cellulose films was similar to the one herein employed, resulted in TS values of 0.217 to 0.882 Mpa, respectively for the control and the calcium-laden films. The inclusion of calcium ions in the structure promoted an increase of 350 % in the tensile strength of the films. Research carried out on films of polysaccharides from spent coffee grounds, employing zinc chloride aqueous solution as solvent, reported a TS value of 0.3 Mpa [29]. The tensile strength of conventional petroleum-derived low-density polyethylene and polyvinyl chloride films used for food packaging are in the range of 10 to 55 Mpa [87]. Scientifically speaking, films prepared from distinct precursor materials and distinct processes should not be compared amongst themselves. Nevertheless, it is plausible to do so when we talk about applicability of films based on a single property, such as tensile strength, i.e., based solely on tensile strength, the films herein produced could be used for the same types of applications that the conventional petroleum-derived films are applied to. However, when all the different properties needed for defining a suitable application are considered, such comparison cannot be straightforwardly made.”

Reviewer 4 Report

The authors  reported a study of using the biomass extracted from coffee beans to obtain films with various croslinking degrees. The authors need to provide additional information so that the study to be clearly understood by a large class of readers. Please find below some comments/suggestions/ advices which may contribute to the increasing the quality of the manuscript.

1. The authors reported some extraction method from the coffee. However, it was not clear enough what compound was extracted and how was it identified to confirm the outcome of the extraction process. The authors are requested to provide additional information. The yield of extraction should be also included.

2. The authors described FT-IR analysis as one of the method to characterize the obtained films. The authors stated “The F511 films were not analyzed by FTIR  because they showed a highly brittle appearance, making the analysis performed with the ATR accessory unfeasible.” However, attenuated total reflection is a sampling technique used in conjunction with infrared spectroscopy (ATR-IR) which enables samples to be examined directly in the solid or liquid state without further preparation. The authors are requested to provide spectra or to insert an explanation; additionally, they may supply another characterization method. The aim of this analysis was not clear enough.

3. The authors discussed within thermodegradation study about cellulose-based material; however, from the study is not clear enough the composition expected, so that to reproduce exactly the assignments of the peaks. The authors are requested to provide additional details.

4. There are a couple of types of crosslinking techniques presented; the motivation of distinctly crosslinking  was not clear enough so there is a need for additional explanation.

5. Conclusion part should be reformulated according to the new information to be included.

Author Response

The authors reported a study of using the biomass extracted from coffee beans to obtain films with various croslinking degrees. The authors need to provide additional information so that the study to be clearly understood by a large class of readers. Please find below some comments/suggestions/ advices which may contribute to the increasing the quality of the manuscript.

  1. The authors reported some extraction method from the coffee. However, it was not clear enough what compound was extracted and how was it identified to confirm the outcome of the extraction process. The authors are requested to provide additional information. The yield of extraction should be also included.

Modified as requested.

  1. The authors described FT-IR analysis as one of the method to characterize the obtained films. The authors stated “The F511 films were not analyzed by FTIR because they showed a highly brittle appearance, making the analysis performed with the ATR accessory unfeasible.” However, attenuated total reflection is a sampling technique used in conjunction with infrared spectroscopy (ATR-IR) which enables samples to be examined directly in the solid or liquid state without further preparation. The authors are requested to provide spectra or to insert an explanation; additionally, they may supply another characterization method. The aim of this analysis was not clear enough.

The way the original text was written in the manuscript was misleading and it was modified to clarify the statement regarding the analysis by FTIR not being feasible. In order to make such statement clearer, the following text replaced the original one in the manuscript: “F511 films were not analyzed by FTIR because they were extremely brittle, rendering their handling virtually impractical. The film crumbled into significantly small pieces upon handling, therefore, making the analysis performed with the ATR accessory unfeasible, since the accessory has a rather large surface area that needs to be entirely covered for the analysis to be suitably carried out.”

The aim of employing this type of analysis is to provide support information to the results of the other techniques by elucidating the presence of specific chemical functionalities that assure the correct characterization of the samples’ composition was duly performed. Also, such analysis provides further information that will help interpretation of the results for the properties that are dependent on the types of compounds present (mostly polysaccharides) and on the nature of the chemical functionalities therein present. Specific functionalities can impart distinct properties to a material, which in turn can affect how this material interacts with other compounds, such as water, acids and bases.

  1. The authors discussed within thermodegradation study about cellulose-based material; however, from the study is not clear enough the composition expected, so that to reproduce exactly the assignments of the peaks. The authors are requested to provide additional details.

To clarify such an issue, data on spent coffee grounds and on the alkaline hydrogen peroxide treated SCG were added to the manuscript. Based on data from the literature and on the information obtained from the FTIR and from the monosaccharide composition analysis, the inference of the types of polysaccharides present in the samples is rather straightforward. Both the spent coffee grounds and the treated samples are comprised mostly by cellulose, arabinogalactose and galactomannans, with the spent coffee grounds presenting higher contents of phenolic compounds. Given the nature of each type of polysaccharide, peaks in the thermograms can also be straightforwardly attributed. The added compounds, such as glycerol, zinc chloride and the crosslinking agents, were duly considered in the attributions.

  1. There are a couple of types of crosslinking techniques presented; the motivation of distinctly crosslinking was not clear enough so there is a need for additional explanation.

Modified as requested. The justification/rationale for using the two distinct types of crosslinking agents was added at the end of the Introduction section, following the statement of the objective of the study.

  1. Conclusion part should be reformulated according to the new information to be included.

Modified as requested.

Round 2

Reviewer 3 Report

The authors have responded to the comments in an acceptable manner.

The language is readable and acceptable. Minor language improvements can help improve the paper further.

Reviewer 4 Report

The authors answered to the addressed queries and the manuscript has been updated